# Enhanced CRISPR-based DNA demethylation by Casilio-ME-mediated RNA-guided coupling of methylcytosine oxidation and DNA repair pathways

Aziz Taghbalout[1], Menghan Du[1], Nathaniel Jillette[1], Wojciech Rosikiewicz[1], Abhijit Rath[2], Christopher D. Heinen[2], Sheng Li[1] & Albert W. Cheng [1,3,4]*

Here we develop a methylation editing toolbox, *Casilio-ME*, that enables not only RNA-guided methylcytosine editing by targeting TET1 to genomic sites, but also by co-delivering TET1 and protein factors that couple methylcytosine oxidation to DNA repair activities, and/or promote TET1 to achieve enhanced activation of methylation-silenced genes. Delivery of TET1 activity by *Casilio-ME1* robustly alters the CpG methylation landscape of promoter regions and activates methylation-silenced genes. We augment *Casilio-ME1* to simultaneously deliver the TET1-catalytic domain and GADD45A (*Casilio-ME2*) or NEIL2 (*Casilio-ME3*) to streamline removal of oxidized cytosine intermediates to enhance activation of targeted genes. Using two-in-one effectors or modular effectors, *Casilio-ME2* and *Casilio-ME3* remarkably boost gene activation and methylcytosine demethylation of targeted loci. We expand the toolbox to enable a stable and expression-inducible system for broader application of the *Casilio-ME* platforms. This work establishes a platform for editing DNA methylation to enable research investigations interrogating DNA methylomes.

[1] The Jackson Laboratory for Genomic Medicine, 10 Discovery Drive, Farmington, CT 06032, USA. [2] Center for Molecular Oncology, University of Connecticut Health, 263 Farmington Avenue, Farmington, CT 06030, USA. [3] Department of Genetics and Genome Sciences, University of Connecticut Health, 400 Farmington Avenue, Farmington, CT 06030, USA. [4] Institute for Systems Genomics, UConn Health Science Center, 400 Farmington Avenue, Farmington, CT 06030, USA. *email: albert.cheng@jax.org

DNA methylation is part of the multifaceted epigenetic modifications of chromatin that shape cellular differentiation, gene expression, and maintenance of cellular homeostasis. Aberrant DNA methylation is implicated in various diseases including cancer, imprinting disorders, and neurological diseases[1]. Developing tools to directly edit the methylation state of a specific genomic locus is of significant importance both for studying the biology of DNA methylation as well as for development of therapies to treat DNA methylation-associated diseases.

In mammalian cells, the 5-methylcytosine (5mC) epigenetic mark generated by covalent linkage of a methyl group to the 5th position of the cytosine ring of CpG sequences is catalyzed by one of the three canonical DNA methyltransferases DMNT1, DNMT3A, and DNMT3B[2–4]. DNA methylation is dynamic and involves demethylation pathways which erase 5mC to restore unmethylated DNA. Active demethylation involves the ten–eleven translocation (TET) family of methylcytosine dioxygenases that iteratively oxidize 5mC into 5-hydroxymethylcytosine (5hmC), 5-formylcytosine (5fC), and 5-carboxylcytosine (5caC) intermediates[5]. Subsequently, 5fC and 5caC are processed by the base-excision repair (BER) machinery to restore unmethylated cytosines. Restoration of an intact DNA base is initiated by DNA glycosylases that excise damaged bases to generate an apurinic/apyrimidinic site (AP site) for processing by the rest of the BER machinery. Thymine DNA glycosylase (TDG)-based BER has been functionally linked to TET1-mediated demethylation, suggesting an interplay between TET1 and enzymes of the BER machinery to actively erase 5mC marks. TDG acts on 5fC and 5caC and NEIL1 and NEIL2 DNA glycosylase/AP-lyase activities facilitate restoration of unmethylated cytosine by displacing TDG from the AP site to create a single strand DNA break substrate for further BER processing[6–12]. Interestingly, DNA demethylation is enhanced by GADD45A (Growth Arrest and DNA-Damage-inducible Alpha), a multifaceted nuclear protein involved in maintenance of genomic stability, DNA repair and suppression of cell growth[13–15]. GADD45A interacts with TET1 and TDG, and was suggested to play a role in coupling 5mC oxidation to DNA repair[16,17].

Advances in artificial transcription factor (ATF) technologies have enabled direct control of gene expression and epigenetic states[18–20]. CRISPR/Cas9-based technologies allow much flexibility and scalability because the specificity is programmable by a single guide RNA (sgRNA)[21,22]. Tethering of TET1 or DNMT3a to genomic targets by use of ATFs has been shown to allow targeted removal or deposition of DNA methylation[23–29]. However, these ATF systems have inherent limitations in enabling multiplexed targeting, effector multimerization or formation of protein complexes at the targeted sequence. We recently developed the Casilio system which uses an extended sgRNA scaffold to assemble protein factors at target sites, enabling multimerization, differential multiplexing[30], and potentially stoichiometric complex formation.

Here we develop an advanced DNA methylation editing technology which allows targeted bridging of TET1 activity to BER machinery to efficiently alter the epigenetic state of CpG targets and activate methylation-silenced genes. Casilio-DNA Methylation Editing (ME) platforms enable targeted delivery of the TET1 effector alone (Casilio-ME1) or in association with GADD45A (Casilio-ME2) or NEIL2 (Casilio-ME3) to achieve enhanced 5mC demethylation and gene activation. We show that Casilio-ME-mediated delivery of TET1 activity to gene promoters induces robust cytosine demethylation within the targeted CpG island (CGI) and activation of gene expression. When systematically compared to other reported methylation editing systems, Casilio-ME shows superior activities in mediating transcriptional activation of methylation-silenced gene and 5mC demethylation. The ability of Casilio-ME to mediate co-delivery of TET1 activity along with other protein factors, which enhance turnover of oxidized cytosine intermediates, paves the way for new areas of research to efficiently address the cause–effect relationships of DNA methylation in normal and pathological processes.

## Results

**Casilio-ME1 delivers TET1 activity to a genomic target site.** Casilio-ME1 is a three-component DNA Methylation Editing platform built on Casilio which uses nuclease-deficient Cas9 (dCas9), an effector module made of Pumilio/FBF (PUF) domain linked to an effector protein, and a modified sgRNA containing PUF-binding sites (PBS) (Fig. 1a)[30]. The dCas9/sgRNA ribonucleoprotein complex binds DNA targets without cutting to serve as an RNA-guided DNA-binding vehicle whose specificity is dictated by the spacer sequence of the sgRNA and a short protospacer adjacent motif located on the target DNA. PUF-tethered effectors are recruited to the ribonucleoprotein complex via binding to their cognate PBS present on the sgRNA scaffold. PUF domains are found in members of an evolutionarily conserved family of eukaryotic RNA-binding proteins whose specificity is encoded within their structural tandem repeats, each of which recognizes a single ribonucleobase[31]. PUF domains can be programmed to bind to any 8-mer RNA sequence, e.g., PUFa and PUFc used in this study were designed to bind PBSa (UGUAUGUA) and PBSc (UUGAUGUA), respectively[30,31]. Multiple PBS added in tandem to the 3′ end of the sgRNA allow concurrent recruitment of multiple PUF-effectors to targeted DNA sequences without interfering with dCas9 targeting, and therefore allow amplification of the response to associated effector modules[30].

To enable targeted cytosine demethylation and subsequent activation of methylation-silenced genes, we built a DNA methyl editor TET1-effector Casilio-ME1 as a protein fusion of hTET1 catalytic domain (TET1(CD)) to the carboxyl end of PUFa (Fig. 1a). We chose as a target the MLH1 promoter region that is part of a large CGI whose aberrant hypermethylation induces MLH1-silencing in 10–30% of colorectal and other cancers[32,33]. MLH1 is silenced in HEK293T cells and therefore represents a clinically relevant model for developing Casilio-ME.

To test the system, cells were transiently transfected with plasmids encoding Casilio-ME1 components PUFa-TET1(CD) effector, dCas9 and six MLH1-promoter-targeting sgRNAs each containing five copies of PBSa (Fig. 1a). This resulted in robust MLH1 activation as indicated by the obtained fold changes in MLH1 mRNA in cells collected on day 3 post-transfection (Fig. 1b, c upper panel). In contrast, MLH1 activation was not obtained with a non-targeting sgRNA (NT-sgRNA) (Fig. 1b), indicating that Casilio-ME1-mediated MLH1 activation requires specific targeting of the PUFa-TET1(CD) module directed by the programmable sgRNAs.

Evidence that the Casilio-ME1-induced activation of MLH1 results from TET1-mediated 5mC demethylation came from high throughput bisulfite sequencing (BSeq) of MLH1 amplicons derived from the cells analyzed in Fig. 1b. BSeq showed that targeted delivery of the TET1(CD) effector induces a profound decrease in CpG methylation frequency within the MLH1 promoter region (Fig. 1c lower panel). Demethylation activity was prominently higher within CpGs neighboring MLH1-sgRNA sites (Fig. 1c lower panel (arrows)), and seemed to spread away, albeit with relatively reduced activities. These data indicate that Casilio-ME1 mediates delivery of TET1

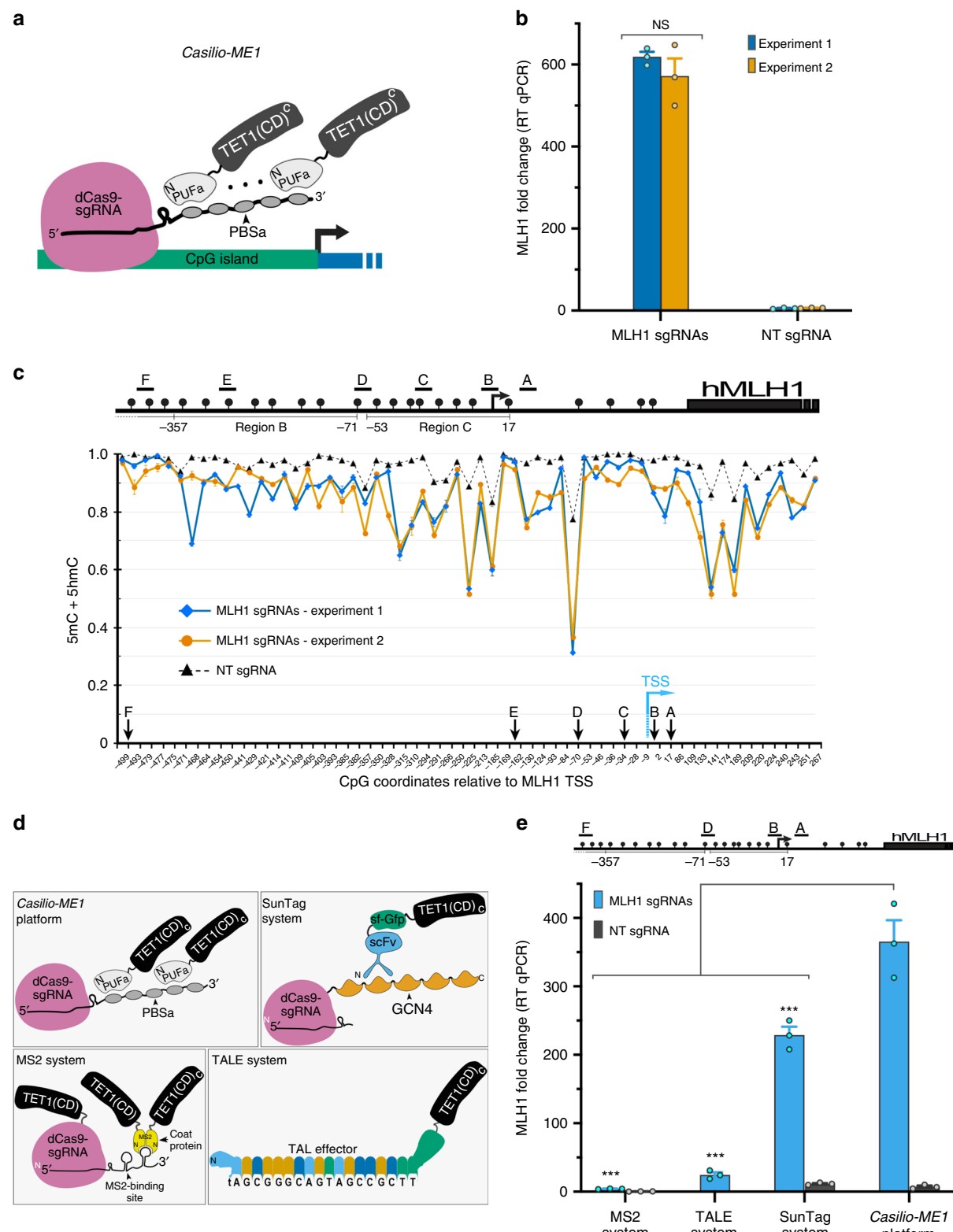

activity to promoter regions to induce 5mC demethylation within the targeted CGI and subsequent activation of the methylation-silenced gene.

**Comparison of *Casilio-ME1* with other TET1 delivery systems.** Although other technologies enabling targeted delivery of TET1 activity to genomic loci have been reported to induce activation of

methylation-silenced genes[23,24,26,29], a direct comparison of their efficiency is lacking. Here we compared *Casilio-ME1* efficiency to alter expression of methylation-regulated genes to alternative technologies for 5mC demethylation that are based on TALEs (transcription activator like effector), dCas9/MS2 or dCas9/Sun-Tag systems[23,24,26] (Fig. 1d). We therefore assembled four TALE-TET1(CD) fusions each designed to bind to one of the four *MLH1*-sgRNA target sequences used for dCas9-based delivery

**Fig. 1** Evaluation of *Casilio-ME1*-mediated gene activation and 5mC demethylation. **a** Schematic representation *Casilio-ME1* components. PUFa-TET1(CD) effector (TET1 residues 1418–2136), dCas9, and sgRNA with 3′extension scaffold comprising five PUFa-binding site (PBSa) are shown. Amino (N) and carboxyl (C) termini of protein fusions are arbitrarily shown. **b** Column plot showing fold changes in *MLH1* mRNA levels in cells transfected with *Casilio-ME1* components comprising *MLH1*-sgRNAs or NT-sgRNA. Cells were collected three days after transfection and were not subjected to selection. Error bars represent mean ± S.E.M ($n = 3$), data form two independent experiments are shown. NS, not significant, $p > 0.05$, two-way ANOVA. **c** Upper panel: *MLH1* promoter and associated CpG island. Regions B and C are depicted according to a report correlating *MLH1*-silencing to region C hypermethylation[33]. CpGs (lollipops), transcription start site (TSS) (arrow), and the sgRNAs used (A to F) are shown. Coordinates are relative to annotated TSS. Lower panel: high throughput BSeq analysis of *MLH1* amplicons obtained from cells analyzed in (b). CpG methylation frequency of *MLH1* promoter regions (mean ± S.E.M; $n = 2$) is shown. Arrows indicate locations of CpG overlapping the *MLH1* sgRNAs (A–F) target sequences or TSS (blue arrow) as shown. CpG coordinates represent positions of cytosines, in base pair, relative to annotated TSS. $p < 0.0001$, two-tail $t$-test and two-way ANOVA. **d** DNA demethylation technologies compared in panel (e). TET1 effectors are tethered to dCas9 nucleoprotein at targeted site via binding of PUFa to PBSa, MS2 coat protein to stem-loop RNA structures appended to sgRNA, or ScFv (single-chain fragment variable) antibody against short peptide (GCN4) appended in array to dCas9 carboxy-terminus. TALE mediates delivery of TET1 activity via binding to targeted sequence. In MS2 system mouse TET1(CD) is also C-terminally fused to dCas9. **e** Evaluation of *Casilio-ME1* platform as compared with alternative 5mC demethylation systems. *MLH1* mRNA relative levels (mean ± S.E.M.; $n = 3$) in cells transfected with *Casilio-ME1*, MS2, TALE or SunTag components. Four *MLH1*-sgRNAs or NT-sgRNA were used with dCas9-based delivery systems. Four TALE effectors each targeting the sequences targeted by the *MLH1*-sgRNAs A, B, D, or F were used. ***$p < 0.05$, one-way ANOVA

systems. Relative quantitation of *MLH1* mRNA indicated that the SunTag, TALEs or MS2 based systems only achieved 63%, 7% or 1%, respectively, of the *Casilio-ME1*-mediated activation level (Fig. 1e). BSeq analysis of *MLH1* promoter comparing *Casilio-ME1* and SunTag systems showed that *Casilio-ME1* induced stronger demethylation at most of the CpG sites examined (Supplementary Fig. 1a, b). The efficient *MLH1* activation obtained with *Casilio-ME1* delivery of TET1 activity as compared to SunTag is not driven by sgRNA composition as a similar trend was obtained when one sgRNA was used for targeting *MLH1* CGI (Supplementary Fig. 1c). These results suggest that *Casilio-ME1*-mediated delivery of TET1 activity to CGI target enables stronger 5mC demethylation and gene activation compared to published systems.

**Co-delivery of TET1(CD) and GADD45A enhances gene activation.** Active 5mC erasure is a two-step process initiated by TET1-mediated iterative 5mC oxidations followed by base-excision (BER) or nucleotide-excision (NER) repair conversions of oxidized intermediates to cytosines[5,15]. Thus, coupling these two steps could streamline 5mC active erasure to efficiently activate methylation-silenced genes. Because GADD45A promotes TET1 activity and/or could recruit key player(s) of DNA repair[14–17], we sought to augment *Casilio-ME* by constructing an upgraded version *Casilio-ME2* which simultaneously recruits TET1(CD) and GADD45A to bridge 5mC oxidation with DNA repair at a specific genomic locus.

To test *Casilio-ME2* in inducing *MLH1* activation in a comparison with *Casilio-ME1* and our previously reported *Casilio-p65HSF1* activator, we introduced plasmids encoding the protein fusions of PUFa-p65HSF1 (activator), PUFa-TET1(CD) (*Casilio-ME1*), PUFa-GADD45A-TET1(CD) (*Casilio-ME2.1*), or GADD45A-PUFa-TET1(CD) (*Casilio-ME2.2*) (Fig. 2a), along with dCas9 and sgRNAs plasmids into HEK293T cells. *Casilio-ME1*-mediated *MLH1* activation was 46% higher than that obtained with the PUFa-p65HSF1 activator module (Fig. 2b). Interestingly, when GADD45A was added as part of *Casilio-ME2.1* or *Casilio-ME2.2* TET1-effectors, *MLH1* mRNA expression was augmented by 3 and 6-fold, respectively, compared to *Casilio-ME1* (Fig. 2b). This enhanced *MLH1* activation, obtained with GADD45A as part of the TET1 effector modules, does not result from higher expression of *Casilio-ME2* effectors (Supplementary Fig. 2). Thus, coupling of GADD45A and TET1(CD) as a two-in-one effector enhances TET1-mediated activation of methylation-silenced genes.

To obtain further evidence that co-delivery of TET1(CD) and GADD45A effectors to target sites enhances gene activation compared to delivery of TET1(CD) alone, we fused each effector

to a separate PUF protein, i.e., TET1(CD) to PUFa and GADD45A to PUFc, and used sgRNA containing both PBSa and PBSc to tether the respective PUF-fusion to the sgRNA scaffold (Fig. 2c). When *Casilio-ME2.3* (PUFc-GADD45A) or *Casilio-ME2.4* (GADD45A-PUFc) components were introduced to cells, 3- and 6-fold increase in TET1-mediated *MLH1* activation was obtained, respectively (Fig. 2d). However, no *MLH1* expression was detected using *Casilio-ME2.3* and *ME2.4* systems when a catalytically dead TET1(CD) (dTET1(CD)) replaced wild-type TET1(CD), indicating that the observed GADD45A-mediated stimulation of gene activation requires the oxidative activity of TET1 (Fig. 2d). Similarly, no *MLH1* activation was obtained when the GADD45A module of *Casilio-ME2.3* and *ME2.4* systems were introduced into cells without the PUFa-TET1(CD) component (Fig. 2c, d), indicating that expression of GADD45A alone, in the absence of the TET1 module, does not mediate gene activation. In addition, when the sgRNAs contained PBSa but lacked PBSc required for tethering PUFc-associated modules, GADD45A modules failed to stimulate *MLH1* activation (Supplementary Fig. 3). Thus, enhancement of gene activation in *Casilio-ME2* requires co-delivery of GADD45A and TET1-effector modules to the target site.

**Co-delivery of TET1(CD) and BER enzymes.** TDG, NEIL1, and NEIL2 have been functionally linked to active DNA demethylation as they are involved in the initial step of removing oxidized cytosines 5fC and 5caC produced by TET1 activities[6–9,11,12]. Because initiating repair of oxidized cytosines by the BER machinery might be a rate limiting step to TET1-mediated activation of methylation-silenced genes, we reasoned that coupling TET1 activities with DNA glycosylases could facilitate 5mC active erasure and enhance subsequent gene activation.

We therefore linked NEIL1, NEIL2, NEIL3, or TDG to the PUFa-TET1(CD) effector as single-chain protein fusions and looked for potential gains in *Casilio-ME1*-mediated *MLH1* activation. Among these, only NEIL2 fusions showed enhanced activation of *MLH1* expression (Supplementary Fig. 4). *Casilio-ME3.1* and *Casilio-ME3.2*, in which NEIL2 is fused N-terminally to PUFa or between PUFa and TET1(CD) of the PUFa-TET1(CD) effector, respectively, increased *MLH1* activation 4-fold in the presence of *MLH1*-sgRNAs compared to *Casilio-ME1* (Fig. 3a, b). Thus, co-delivering TET1(CD) and NEIL2 as a two-in-one effector to target sites improves activation of 5mC-silenced genes.

To further show that NEIL2 promotes demethylation-mediated gene activation, TET1(CD) and NEIL2 were co-delivered as separate effectors to *MLH1* promoter regions by fusing TET1(CD) and NEIL2 to PUFa and PUFc, respectively, and using

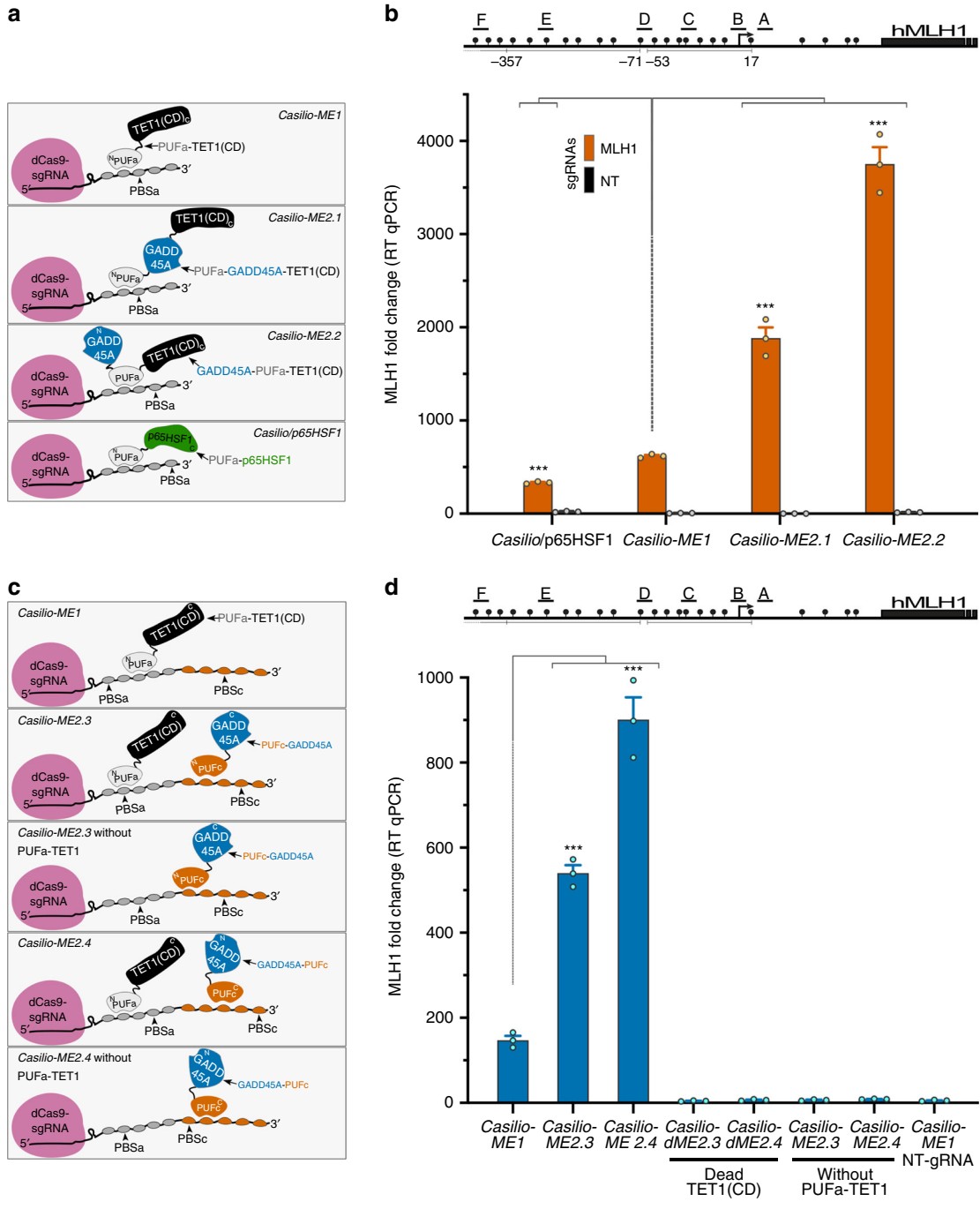

**Fig. 2** *Casilio-ME2* mediates dual delivery of TET1(CD) and GADD45A to targeted genomic sites. **a** Schematic representation of the indicated *Casilio-ME and Casilio* platforms showing effector modules of PUFa fusion proteins used to transfect cells analyzed in (b). TET1(CD) (black), GADD45A (blue), p65HSF1 (green), PUFa (light gray), amino (N), and carboxyl (C) termini of protein fusions and occupancy of PBSa are arbitrarily shown. **b** Plot showing *MLH1* mRNA relative levels (mean fold change ± S.E.M.; $n = 3$) in cells transfected with components of *Casilio-ME1, Casilio-ME2.1, Casilio-ME2.2*, or *Casilio/p65HSF1* in the presence of *MLH1*-sgRNAs or NT-sgRNA as indicated. Cells were collected 3 days after transfection. Drawing of promoter regions with the *MLH1*-sgRNAs used (A–F), CpGs (lollipops), and TSS (arrow) is shown above the plot. ***$p < 0.0005$, one-way ANOVA. **c** Schematic representation of the indicated *Casilio-ME* platforms showing effector modules of PUFa and PUFc fusion proteins used to transfect cells analyzed in panel (d). TET1(CD) (black), GADD45A (blue), PUFa (light gray), PUFc (orange), sgRNA containing both PBSa and PBSc, amino (N) and carboxyl (C) termini of protein fusions are arbitrarily shown. **d** *MLH1* mRNA relative levels (mean fold change ± S.E.M.; $n = 3$) in cells transfected with components of *Casilio-ME1, Casilio-ME2.3*, or *Casilio-ME2.4* in the presence of *MLH1*-sgRNAs or NT-sgRNA is shown. When indicated PUFa-TET1(CD), effector component of *Casilio-ME2.3* and *Casilio-ME2.4*, was replaced by a catalytically dead PUFa-TET1(CD) effector containing TET1-inactivating mutations H1671Y, D1673A[57], or omitted. *MLH1* promoter with the sgRNAs used (A–F), CpGs (lollipops), and TSS (arrow) is depicted above the plot. ***$p < 0.0005$, one-way ANOVA

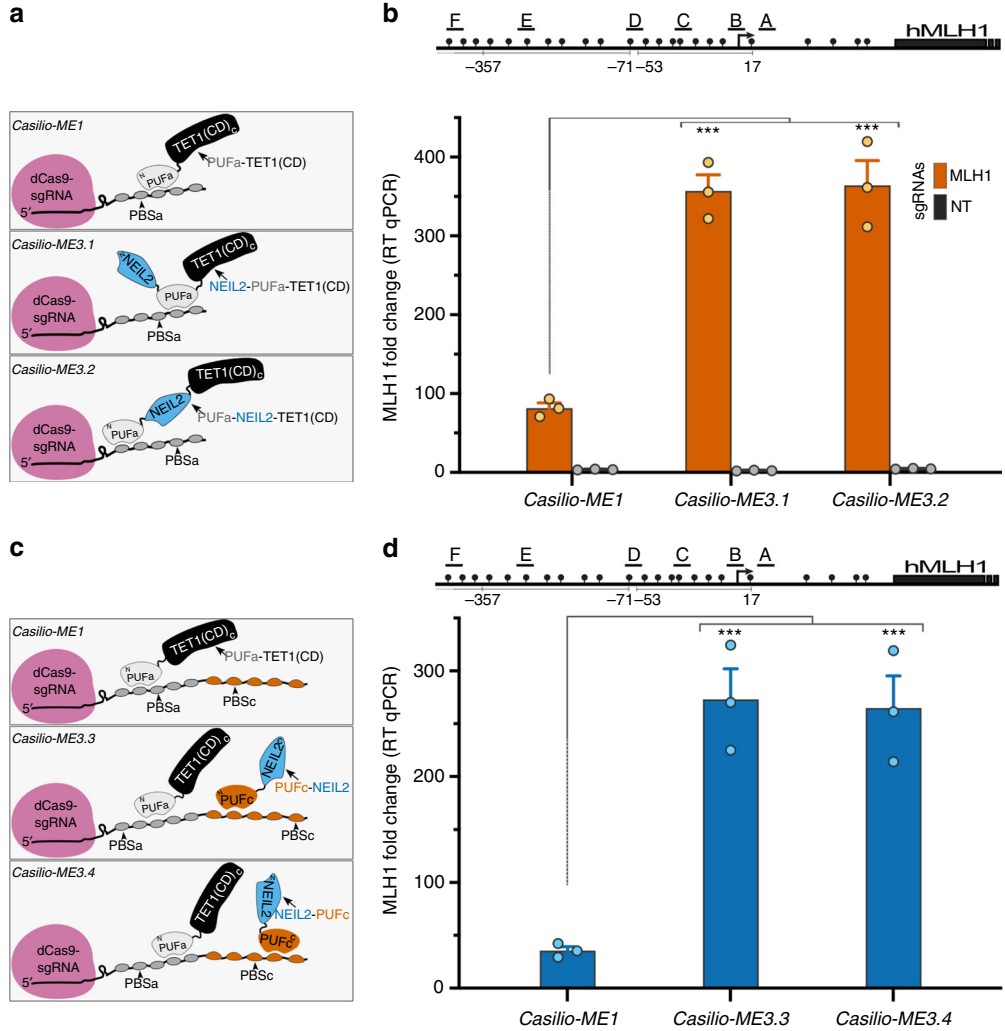

**Fig. 3** *Casilio-ME3* mediates dual delivery of TET1(CD) and NEIL2 to targeted genomic site. **a** Illustration of the indicated *Casilio-ME* platforms showing effector modules of PUFa fusion proteins used to transfect cells analyzed in panel (b). TET1(CD) (black), NEIL2 (blue), PUFa (light gray), occupancy of PBSa, amino (N) and carboxyl (C) termini of protein fusions are arbitrarily shown. **b** *MLH1* mRNA relative levels (mean fold change ± S.E.M.; $n = 3$) in cells transfected with components of *Casilio-ME1, Casilio-ME3.1,* or *Casilio-ME3.2* in the presence *MLH1*-sgRNAs or NT-sgRNA as indicated. Drawing of promoter regions with the *MLH1*-sgRNAs used (A–F), CpGs (lollipops), and TSS (arrow) is shown above the plot. ***$p < 0.001$, one-way ANOVA. **c** Schematic representation of the indicated *Casilio-ME* platforms showing effector modules of PUFa and PUFc protein fusions used to transfect cells analyzed in panel (d). TET1(CD) (black), NEIL2 (blue), PUFa (light gray), PUFc (orange), sgRNA containing both PBSa and PBSc, occupancy of PBSa and PBSc, amino (N) and carboxyl (C) termini of protein fusions are arbitrarily shown. **d** Plot showing *MLH1* mRNA relative levels (mean fold change ± S.E.M.; $n = 3$) in cells transfected with components of *Casilio-ME1, Casilio-ME3.3,* or *Casilio-ME3.4* in the presence of *MLH1*-sgRNAs. Drawing of promoter regions with the *MLH1*-sgRNAs used (A–F), CpGs (lollipops), and TSS (arrow) is shown above the plot. ***$p < 0.005$, one-way ANOVA

sgRNAs comprising both PBSa and PBSc (Fig. 3c). Binding of these effectors to a sgRNA scaffold brings TET1(CD) and NEIL2 into close proximity, and potentially enables coupling of DNA demethylation with BER. When *Casilio-ME3.3* (PUFc-NEIL2) or *Casilio-ME3.4* (NEIL2-PUFc) components were used, *MLH1* activation was increased by 7-fold as compared to *Casilio-ME1* (Fig. 3d). Taken together, these results show that co-delivery of TET1(CD) and NEIL2 DNA glycosylase/AP-lyase stimulates activation of a methylation-silenced gene.

To determine whether the enhanced gene activation obtained with *Casilio-ME3.3 and ME3.4* systems requires co-targeting of TET1(CD) and NEIL2 to a genomic site and does not result from NEIL2 over expression, we disabled targeting of NEIL2 effector modules by using sgRNAs comprising PBSa but lacking the PBSc required for targeting PUFc-based NEIL2 effectors (Supplementary Fig. 5a). Cells transfected with *Casilio-ME3.3 or ME3.4* components comprising sgRNAs that lacked PBSc tethering sites

showed no significant gains in TET1-mediated *MLH1* activation, indicating that enhanced TET1-mediated gene activation requires co-targeting of NEIL2 and TET1(CD) modules via an RNA scaffold (Supplementary Fig. 5b). Thus, these data show that co-delivery of NEIL2 and TET1(CD) to genomic loci synergistically promotes TET1-mediated gene activation, likely via facilitated coupling of 5mC demethylation and BER activities to efficiently restore unmethylated cytosine to targeted sites.

**Comparison of *Casilio-ME* platforms.** *Casilio-ME2* and *Casilio-ME3* platforms showed an enhanced activation of a methylation-silenced gene compared to *Casilio-ME1*. Here we sought to compare these platforms to one another in their efficiencies to activate *MLH1* and alter methylation landscape of targeted CGI. The comparison included the previously reported dCas9-TET1 as an alternative system for reference[25]. Normalized *MLH1* levels, to those obtained with dCas9-TET1 system, showed that *Casilio-*

ME2.2 gave augmented *MLH1* activation higher than *Casilio-ME1, ME2.1, ME3.1*, and *ME3.2* platforms (Supplementary Fig. 6a). This augmented *MLH1* activation does not necessarily require targeting with multiple sgRNAs as a similar trend in *MLH1* activation was obtained with one sgRNA targeting *MLH1* promoter region (Fig. 6Sb). When modular *Casilio-ME2.3, ME2.4*, and *ME3.4* platforms were compared, *Casilio-ME2.4* showed the most *MLH1* activation (Supplementary Fig. 6c). Only background *MLH1* mRNA levels could be detected with TET1 (CD)-dead mutants of the *Casilio-ME* platforms (*Casilio-dME*) (Supplementary Fig. 6a, c), indicating that TET1 activity is required for gene activation and that delivery of GADD45A or NEIL2 without TET1 oxidative activity is not sufficient for activating methylation-silenced genes.

To ask whether the augmented *MLH1* activation of *Casilio-ME2.2* and *Casilio-ME3.1* came from an increased efficiency in 5mC erasure, we performed BSeq and oxidative BSeq (oxBSeq) by high throughput amplicon sequencing of *MLH1* promoter regions derived from cells transfected with *Casilio-ME* components or dCas9-TET1. Analysis of 5mC frequencies within *MLH1* CGI showed that *Casilio-ME2.2* and *Casilio-ME3.1* produced higher demethylation activities compared to *Casilio-ME1* and dCas9-TET1 as indicated by the reduced levels of 5mC (Fig. 4a upper panel, b). This is consistent with the observed higher accumulation trends of 5mC-oxidation products 5hmC and bisulfite converted CpGs (5fC, 5caC and C) (Fig. 4). Interestingly, a noticeable trend appears to exist when looking at the levels of 5mC-oxidation products; *Casilio-ME2.2* produced more bisulfite converted CpGs (5fC, 5caC, and C), whereas *Casilio-ME3.1* produced more 5hmC (Fig. 4b, c). This apparent difference in 5mC oxidation patterns could explain the relative efficiencies of *Casilio-ME2.2* and *ME3.1* in enhancing *MLH1* activation. The higher accumulation of 5hmC in the NEIL2-based *Casilio-ME* platform could be explained by NEIL2 competing with TET1 (CD) for processing 5fC and 5caC substrates to potentially steer TET1 activity more toward the 5mC substrate. Alternatively, TET1 activity may be promoted in the presence of NEIL2 or NEIL2-associated proteins. For *Casilio-ME2.2*, the observed 5mC oxidation profiles are consistent with GADD45A promoting TET1 activity and/or recruiting BER to the target site, leading to accumulation of bisulfite converted CpGs (5fC, 5caC, and C).

Evidence that the enhanced gene activation obtained with *Casilio-ME2.2* and *Casilio-ME3.1* required fully active GADD45A or NEIL2, respectively, was obtained when point mutations were introduced to alter key functional features or inactivate corresponding proteins. GADD45A lacks any obvious enzymatic activity; however, previous reports pointed us to key amino acids required for chromatin interaction (G39A) or dimerization/self-association (L77E) of the protein[34,35]. Catalytically inactive NEIL2 with (C291S) or (R310Q) mutations located at the zinc finger domain required for NEIL2-binding to DNA substrate were also reported[36]. Introduction of these point mutations to *Casilio-ME2.2* or *Casilio-ME3.1* abrogated the enhanced *MLH1* activation (Supplementary Fig. 7a). The reduced *MLH1* activations obtained were not due to protein destabilization caused by amino-acid changes introduced to GADD45A and NEIL2 (Supplementary Fig. 7b, c). Interestingly, *Casilio-ME3.1* containing NEIL2(R310Q) mutation seemed to retain a weak enhancement that is likely attributed to residual catalytic and DNA-binding activities of the R310Q NEIL2 mutant (Supplementary Fig. 7a)[36].

The enhanced demethylation activities, taken together with the fact that the boost in *MLH1* activation mediated by *Casilio-ME2.2* and *Casilio-ME3.1* required TET1 oxidative activity and functionally active GADD45A or NEIL2 enhancer proteins, is consistent with the idea that these platforms might facilitate

bridging oxidative removal of 5mC to DNA repair pathways to efficiently restore unmethylated cytosine to targeted loci.

**Evaluation of potential off-target activities of *Casilio-ME*.** The CRISPR/dCas9 system inherently tolerates mismatches, to some extent, between guide RNAs and genomic loci to subsequently give rise to potential off-target effects[37,38]. To evaluate *Casilio-ME* platforms for potential off-target effects, we performed reduced representation bisulfite sequencing (RRBS) of genomic DNA extracted from cells transfected with *Casilio-ME* components, dCas9-TET1 or SunTag systems in the presence of either *MLH1* or non-targeting sgRNAs. Pairwise correlations between all samples, including untransfected cells, gave similar correlations. The correlations were within the same range as previously reported for RRBS replicates[39], suggesting that *Casilio-ME* platform associated off-target activities, if any existed, do not exceed those of alternative 5mC editing systems (Supplementary Fig. 8a).

To evaluate further the specificity of the *Casilio-ME* platforms, we performed RNAseq to compare *MLH1*-sgRNAs and NT-sgRNA transfected cells. The overall pattern of gene expression of *Casilio-ME2.2*, SunTag and dCas9-TET1 systems seemed largely similar with high correlations of FPKM values among *MLH1*-sgRNA and NT-sgRNA transfected cells in each system (Supplementary Fig. 8b). In addition to *MLH1*, *FSBP* was called significantly upregulated in *Casilio-ME2.2* by RNAseq analysis, thus representing a potential off-target effect. However, quantitation by TaqMan assays of *FSBP* levels in the RNA samples used for RNAseq showed no expression changes, indicating that *FSBP* activation observed in RNAseq is a false positive (Supplementary Fig. 9a). *MLH1* was more prominently upregulated in *Casilio-ME2.2* transfected cells. The other differentially expressed RNAseq hits of *Casilio-ME2.2*, SunTag and dCas9-TET1 systems could represent off-target effects or reflect potential transcriptome changes subsequent to *MLH1* reactivation (Supplementary Fig. 8b).

Recruitment of BER associated proteins via *Casilio-ME* platforms might introduce mutations to targeted sites. To evaluate potential mutagenicity of *Casilio-ME*, we performed deep sequencing of the *MLH1* locus, a 1 kb targeted region that comprises the promoter and part of the first exon, and compared sequence identity distribution among reads to untransfected cells. *Casilio-ME* transfected cells showed no significant difference in sequence identity within *MLH1* reads, ruling out the possibility of *Casilio-ME* platforms introducing mutations to targeted sites (Supplementary Fig. 9b, c).

To ask whether expression of the PUFa-TET1(CD) fusion proteins, *Casilio-ME* components comprising GADD45A or NEIL2, could cause cellular alterations, we performed proliferation assays (MTT) on transfected cells. Cells passaged after transfection with *Casilio-ME* showed no measurable growth changes compared to controls or cells transfected with components of alternative DNA demethylation systems (Supplementary Fig. 10a).

To further evaluate potential alteration of transcriptomes that may occur subsequent to expression of GADD45A or NEIL2 as part of PUFa fusions, we compared RNA expression profiles in cells transfected with *Casilio-ME1*, *Casilio-ME2.2* or *Casilio-ME3.1*. RNAseq analysis showed overall comparable RNA expression profiles in these cells with high correlations of FPKM values (Supplementary Fig. 10b). However, few off-target genes were called significantly upregulated in the case of *Casilio-ME 2.2* (Supplementary Fig. 10b). None of the off-target hits called genes associated with known GADD45A cellular functions. TaqMan assays on four upregulated off-target genes, HSPA1A, PKP3, SRPX2, and THBS2, showed an activated expression of these off-

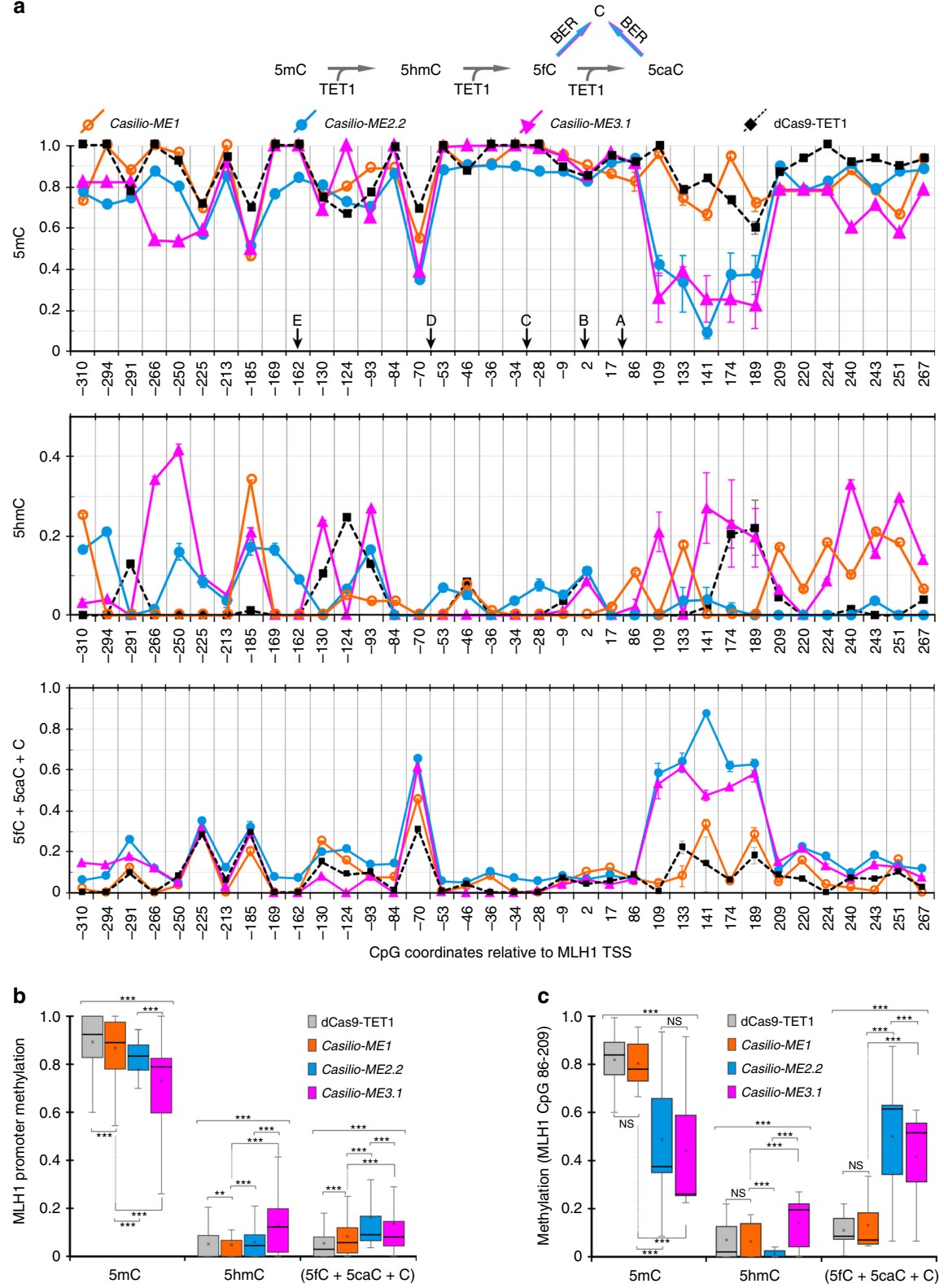

target genes in *Casilio-ME2.2* transfected cells (Supplementary Fig. 10c). Nonetheless, the activated expression of the four genes required TET1 activity and fully functional GADD45A, indicating that expression of GADD45A as part of PUFa-TET1 (CD) protein fusion is not sufficient for inducing the observed changes in expression of untargeted genes (Supplementary Fig. 10c). The CRISPR associated off-target activity could likely

be reduced by using dCas9 variants with higher fidelity as a delivery vehicle[40–43].

**Portability of the *Casilio-ME* platforms**. To show that *Casilio-ME* platforms enable efficient activation of other 5mC-silenced genes and in different cell types, we measured changes in

**Fig. 4** Efficiency of 5mC demethylation induced by *Casilio-ME* platforms and dCas9-TET1 system. *MLH1* promoter was targeted by using components of the indicated methylation editing system in the presence of *MLH1*-sgRNAs. Cells were collected 3 days after transfection and corresponding genomic DNA was subjected to high throughput amplicon BSeq and oxBSeq. **a** 5mC conversion to cytosine by TET1 and BER pathways is depicted above panels. Frequencies (mean ± S.E.M.; $n = 2$) of 5mC (upper panel), 5hmC (middle panel) and bisulfite converted CpG (C, 5fC, and 5caC) (lower panel) plotted against CpG positions within *MLH1* promoter region are shown. The obtained levels of (5mC + 5hmC) and 5mC from two experiments were concordant. Arrows indicate locations of CpG overlapping one of the six sgRNAs target sequences. Statistical significance of differences in methylation patterns were tested. $p < 0.0001$, two-way ANOVA. **b** Box plot of frequencies of different CpG variants measured by BSeq and oxBSeq across the *MLH1* promoter regions in cells transfected with the indicated 5mC demethylation system is shown. For each box plot the thick line inside the box represents the median value and the surrounding bottom and top lines represent the 25th and 75th percentiles. The whiskers represent min and max values, the $x$ represents the mean value. $**p < 0.01$, and $***p < 0.0001$, two-way ANOVA. **c** same as **b** but focusing on CpG 86–209 of the *MLH1* promoter proximal-intron1 region. NS, not significant $p > 0.05$, and $***p < 0.01$, two-way ANOVA

expression of *MGMT*, *SOX17*, *RHOXF2* (HEK293T), *CDH1* (U2OS), and *GSTP1* (LNCaP) in cells transfected with the components of *Casilio-ME* platforms. This showed that *Casilio-ME* platforms also enabled activation of these methylation-silenced genes. *Casilio-ME2.2*, *Casilio-ME3.1*, or *ME3.2* produced significantly enhanced gene activations compared to *Caslio-ME1* for the genes tested (Supplementary Fig. 11a-e). The improved gene activation is consistent with the increased demethylation efficiency obtained by *Casilio-ME3.2* targeting *GSTP1* CGI as compared to *Casilio-ME1* (Supplementary Fig. 11g, h). Expression of PUFa protein fusions of *Casilio-dME1*, *dME2.2*, and *dME3.1*, containing catalytically inactive TET1(CD), in the absence of sgRNAs failed to activate MGMT, SOX17, and RHOXF2 (Supplementary Fig. 11f), indicating that expression of GADD45A or NEIL2 as part of PUFa-TET1(CD) is not sufficient for activating expression of the tested genes.

Interestingly, the superiority and the levels of enhancement in TET1-mediated gene activation achieved by *Casilio-ME2.2* and *Casilio-ME3.1 or ME3.2* varied for some gene targets, suggesting the existence of some locus dependency for GADD45A or NEIL2 to efficiently augment TET1-mediated gene activation. Nonetheless, activation of methylation-silenced genes obtained by *Casilio-ME1* targeting different CGIs in different cell types was more effective compared to previously reported methods (Supplementary Fig. 11i).

**Inducible *Casilio-ME* platform.** To enable tunable and "on-demand" targeted DNA demethylation and gene activation, we constructed piggyBac (PB) transposon vectors hosting doxycycline-inducible PB *Casilio-ME1* cassettes (DIP_*Casilio-ME1*) where the expression of dCas9 and PUFa-TET1(CD) is under the control of Tet-On promoters (Supplementary Fig. 12a). A DIP_*Casilio-ME1* stable cell line is established by piggyBac transposition followed by antibiotic selection. When the DIP_*Casilio-ME1* cell line was transiently transfected with targeting sgRNAs, we obtained robust *MLH1* activation in the presence of doxycycline. *MLH1* mRNA level also increased in response to increasing amounts of doxycycline (Supplementary Fig. 12b). Without doxycycline added, only background levels of *MLH1* were detected and no detectable amounts of *Casilio-ME1* protein components were observed in Western blot analysis of protein extracts from transfected cells (Supplementary Fig. 12b, c). This DIP_*Casilio-ME1* will enable establishment of isogenic cell lines that can be used to study different target CGIs in a tunable manner by supplying different target-specific sgRNAs and adjusting doxycycline dosage.

## Discussion

This study establishes a modular RNA-guided DNA methylation editing platform that not only recruits the TET1 effector to initiate DNA demethylation by 5mC oxidations, but also delivers

protein factors to facilitate coupling 5mC oxidation to DNA repair pathways to effectively restore intact DNA to targeted sites. Such dual delivery enhanced 5mC demethylation at CGI target and augmented gene activations when compared to TET1(CD) delivered alone. In addition to the robustness of the platform, the modular design of *Casilio-ME* allows a high degree of tunability and flexibility in editing 5mC epigenetic marks.

Turnover of 5fC and 5caC by DNA repair machinery lags behind TET1-mediated 5mC oxidations as these intermediates accumulate before getting converted to unmethylated cytosine[44]. Coupling TET1 activity with BER or NER pathways could accelerate 5fC and 5caC turnover, thereby enhancing activation of methylation-silenced genes. Consistent with this idea, *Casilio-ME2* and *Casilio-ME3* platforms designed to facilitate coupling TET1 and DNA repair activities gave an enhanced gene activation and CpG demethylation of targeted sites. This enhanced gene activation requires TET1 catalytic activity, fully functional GADD45A or NEIL2 proteins and co-targeting relevant effectors in close proximity to genomic target sites.

Previous studies revealed interesting functional and physical interactions among proteins involved in oxidizing 5mC and removal of oxidized cytosine intermediates via BER or NER. NEIL2 promotes substrate turnover by TDG during DNA demethylation[12]. GADD45A physically interacts with TET1 or TDG and seems to promote TET1 activity, and enhances removal of 5fC and 5caC by TDG[14,16,17]. GADD45A also recruits repair enzymes such as the 3′-NER endonuclease XPG to genomic sites DNA[45,46]. As GADD45A is devoid of any enzymatic activity, it was proposed to function as a liaison protein to physically couple 5mC oxidation with DNA repair[16]. Consistent with these observations, *Casilio*-mediated co-targeting of TET1(CD) with GADD45A or NEIL2 within close proximity of their substrates enhanced 5mC demethylation and activation of methylation-silenced genes. However, it was reported that GADD45A and TDG failed to enhance demethylation of methylated plasmid by TET1(CD) in vitro[47], and the addition of TDG to *Casilio-ME* modules failed to augment gene activation. TDG protein fusions tested here might not be functional or other factor(s) might be required for TDG to produce enhanced gene activation. The enhanced activation of methylation-silenced genes observed could alternatively be achieved by TDG-independent pathways, by perhaps recruiting yet to be found players that could act by enhancing TET1/BER activities in the presence of GADD45A or NEIL2, or inhibiting DNA methyltransferases leading to replication-dependent demethylation at the targeted sites. The mechanisms by which these potential partners enhance 5mC erasure are in need of further studies. The *Casilio-ME2* and *Casilio-ME3* have the potential to be used in such mechanistic studies as different combinations of protein assemblies that include mutants or other proteins could be tested. Future characterization of protein domains enabling an enhanced TET1-mediated gene activation and of protein interactions taking place

at TET1-targeted genomic sites could lead to further improvement of the *Casilio-ME* platforms and shed light on 5mC editing in mammalian cells.

Different levels of activation of methylation-silenced genes could be obtained by using one of the three flavors of *Casilio-ME* and by varying doxycycline concentrations with the DIP_*Casilio-ME1* platform described here. This equips *Casilio-ME* platforms with the capability to fine-tune gene activation. These *Casilio-ME* platforms significantly expand 5mC editing capability to efficiently address the causal-effect relationships of methylcytosine epigenetic marks in numerous biological and pathological systems.

## Methods

**Cell culture and transfection**. HEK293T and U2OS cells (both from ATCC) were cultivated in Dulbecco's modified Eagle's medium (DMEM)(Sigma) with 10% fetal bovine serum (FBS)(Lonza), 4% Glutamax (Gibco), 1% Sodium Pyruvate (Gibco), and penicillin-streptomycin (Gibco) in an incubator set to 37 °C and 5% $CO_2$. LNCaP cells were obtained from ATCC and cultivated in RPMI-1640 supplemented with 10% FBS. Doxycycline (Dox) (Sigma) (1 μg/ml) or as otherwise indicated was added on the day of transfections with a daily change of media supplemented with Dox. Cells were seeded into 12-well plates at 150,000 cells per well the day before being transfected with plasmids each encoding dCas9 (100 ng), sgRNAs (100 ng) or PUF-fusion (200 ng) in the presence of Attractene or Lipofectamine 3000 transfection reagents according to manufacturers' instructions (Qiagen, Thermo Fisher Scientific). The same plasmid ratio and total amount of DNA were used in cell transfections with components of dCas9/MS2 and SunTag systems. In the two-component systems, where TET1(CD) was fused to dCas9 or TALEs, 200 ng effector, 100 ng sgRNAs, and 100 ng empty vector of plasmid DNA were used. The combinations of *MLH1* sgRNAs used were based on the obtained *Casilio-ME1*-mediated *MLH1* activation efficiency in preliminary experiments. However, no sgRNAs optimizations were performed for the other methylation-silenced genes. Nucleofections of LNCaP cells were performed by using 4D-nucleofector according to manufacturer's instructions (Lonza) using 400 ng plasmid DNA. Cells were harvested 3 days after transfection or as otherwise indicated with media changes at 24 h post-transfection, and cell pellets were used for extractions of RNA, genomic DNA and protein using AllPrep DNA/RNA/Protein Mini Kit according to the manufacturer's instructions (Qiagen). Stable and Dox-inducible expression cell line was generated by transfecting three plasmids including the hyperactive transposase plasmid (hyPBase)[48,49] and the indicated PiggyBac vectors hosting cassette enabling a Dox-inducible expression of dCas9 or PUFa-TET1(CD) effector. The transfected cells were then subjected to double selection in the presence of blasticidin and hygromycin.

**Plasmid constructions**. A list of plasmids with links to their Addgene entries are provided in Supplementary Table 1. Detailed descriptions and sequences of oligonucleotides and proteins are given in the Supplementary Tables 2-5. The plasmids pCAG-dCas9-5xPlat2AfID and pCAG-scFvGCN4sfGFPTET1CD (Addgene #82560 and 82561, respectively), pdCas9-Tet1-CD, and pcDNA3.1-MS2-Tet1-CD (Addgene #83340 and 83341, respectively) were gifts from Izuho Hatada and Rongui Hu, respectively.

**Quantitative RT-PCR analysis**. Harvested cells were washed with Dulbecco's phosphate-buffered saline (dPBS), centrifuged at 125 × g for 5 min and then the flash-frozen pellets were stored at −80 °C. Extracted RNA (500 ng–2 μg) were used as templates to make cDNA libraries using a High Capacity RNA-to-cDNA kit (Applied Biosystems). TaqMan gene expression assays were designed using *GAPDH* (Hs03929097, VIC) as an endogenous control and *CDH1* (Hs01023895_m1, FAM), *GSTP1* (Hs00943350_g1, FAM), HSPA1A (Hs00359163_s1, FAM), *MGMT* (Hs01037698_m1, FAM), *MLH1* (Hs00179866_m1, FAM), PKP3 (Hs00170887_m1, FAM), *RHOXF2* (Hs00261259_m1, FAM), SOX17 (Hs00751752_s1, FAM), SRPX2 (Hs00997580_m1, FAM), or THBS2 (Hs01568063_m1, FAM) as targets (Thermo Fisher Scientific). Quantitative PCR (qPCR) was performed in 10 μL reactions by using TaqMan Universal Master Mix II with UNG and 2 μL of diluted cDNA from each sample (Applied Biosystems). Gene expression levels were calculated by "delta delta Ct" and normalized to control samples using ViiA7 version 1.2.2 or QuantStudio version 1.3 software (Applied Biosystems by Life technologies).

**Bisulfite and oxidative bisulfite sequencing**. Bisulfite and oxidative bisulfite conversion experiments were performed by using the EpiTect Fast DNA Bisulfite Kit, True Methyl oxBS Module and genomic DNA according to manufacturers' instructions (Qiagen and NuGen, respectively). For oxBSeq, oxidation reactions were carried out in parallel and all treated samples developed same orange color expected for successful oxidative reactions. The average bisulfite conversion rates of cytosines in oxidized and non-oxidized sample sets were 0.996 and 0.995,

respectively. Bisulfite treated DNA served as templates to PCR-amplify three DNA fragments of 350–400 bp or a single fragment that cover *MLH1* or *GSTP1* promoter regions, respectively, using ZymoTaq PreMix according to manufacturer's instructions (Zymo Research). The *MLH1* PCR fragments were then cloned by SLIC into *Eco*RI-linearized pUC19 plasmid using T4 DNA polymerize[50]. Ten independent positive clones for each sample were then subjected to Sanger sequencing to determine methylation profiles based on bisulfite-mediated cytosine to thymine conversion frequency of individual CpGs. *MLH1* amplicons obtained from bisulfite converted DNA templates and from unconverted DNA were subjected to high throughput sequencing (2 × 250 paired-end reads) conducted at Genewiz (South Plainfield, NJ, USA). Fifty to 120 thousand reads were obtained per sample. Sequence analysis of the plasmids extracted from *MLH1* clones to determine methylation frequencies was performed by using BiQ Analyzer 3 with minimal bisulfite conversion rate and sequence identity set to 97 and 95%, respectively[51]. Reads from high throughput amplicon sequencing, on the other hand, were analyzed for 5mC and 5hmC by using BiQ Analyzer HiMod with minimal read quality score, alignment score, sequence identity and bisulfite conversion rate set to 30, 1000, 0.9, and 0.9, respectively[52]. Obtained levels of (5mC + 5hmC) and 5mC from two experiments were concordant across the CpGs. BiQ Analyzer HiMod was used without sequence identity filter to analyze sequence integrity of *MLH1* amplicons derived from genomic DNA without bisulfite treatment.

**Reduced representation bisulfite sequencing (RRBS)**. Library preparation for RRBS was performed according to manufacturer's instructions (Diagenode). Briefly, 100 ng of genomic DNA for each sample was enzymatically digested, end-repaired and ligated with an adaptor. Samples with different adaptors were then pooled together and subjected to bisulfite treatment followed by purification steps. The pooled DNA was PCR-amplified and then cleaned up with Ampure XP beads. Libraries were quantified with real time qPCR and sequenced using Illumina NextSeq (1 × 75 single end reads). Forty to 60 million reads per sample were obtained. To compute the CpG methylation levels, RRBS reads were aligned to the hg38 human reference genome using Bismark (version 0.16.0)[53]. CpG sites with coverage lower than 10 or higher than 400 reads were filtered out and Pearson's correlation of methylation profiles across indicated samples were computed by using methylKit (version 1.0.0)[54].

**RNA sequencing**. RNA sequencing libraries were prepared for independent replicates of mRNA samples at Genewiz (South Plainfield, NJ, USA) by using NEBNext Ultra RNA Library Prep Kit for Illumina according to manufacturer's instructions (New England Biolabs). Briefly, mRNA was enriched with Oligod(T) beads, fragmented for 15 min at 94 °C and then reverse transcribed followed by second strand cDNA synthesis. cDNA fragments were end repaired and adenylated at 3′ends, and universal adapters were ligated to cDNA fragments, followed by index addition and library enrichment by limited cycle PCR. The libraries were validated on the Agilent TapeStation (Agilent Technologies), and quantified by using Qubit 2.0 Fluorometer (Invitrogen) and quantitative PCR (KAPA Biosystems). The libraries were clustered on two lanes of a flowcell then loaded on the Illumina HiSeq instrument according to manufacturer's instructions. The samples were sequenced using a 2 × 150 paired-end configuration and image analysis and base calling were conducted by the HiSeq Control Software. Generated raw sequence data (.bcl files) from Illumina HiSeq was converted into fastq files and de-multiplexed using Illumina's bcl2fastq 2.17 software. Thirty to 40 million read pairs were obtained per sample. Reads were then quantitated by Salmon into transcript estimates[55], then subjected to DESeq2 for differential gene expression analysis[56]. Genes with low read counts in all samples were filtered out to eliminate noise in the analysis. Differentially expressed genes were called with adjusted *p*-values less than 0.05. MA-plots and FPKM scatter plots were generated using DESeq2.

**Western blot analysis**. Protein cell extracts (30 μg) were separated by electrophoresis on 10% SDS-polyacrylamide gels and then transferred to nitrocellulose membranes at 100 V for 1 h using Bjerrum Schafer-Nielsen buffer with SDS. Blocked membrane in 5% Blotting-Grade Blocker (BioRad) in TBS-T (50 mM Tris pH 7.6, 200 mM NaCl, 0.1% Tween 20) were incubated overnight at 4 °C with the indicated antibodies, and then protein bands were detected using Horseradish peroxidase-conjugated secondary antibodies (Sigma) and Clarity Western ECL Substrate according to manufacturer's instructions (BioRad). Monoclonal anti-Flag (dilution 1:1000, Sigma cat #F1804), monoclonal CRISPR/Cas9 (dilution 1:5000, Epigentek cat #A-9000-100) and monoclonal anti-β-actin (dilution 1:5000, Sigma cat #MAB 1501) antibodies were used according to manufacturers' instructions. Blots were imaged using a G:Box (Syngene). Uncropped and unprocessed images of the blots are supplied in Source Data file.

**Cell proliferation assay**. Cells were split 48 h after transfection in serial dilutions as indicated, incubated for 48 h and then MTT assay was performed according to manufacturer's instructions (Thermo Fisher Scientific), and absorbance at 540 nm was recorded using a SpectraMax M5 plate reader (Molecular devices).

**Statistical analyses**. Information on replication, statistical tests and presentation are given in the respective figure legends. GraphPad Prism 8 and Microsoft Excel were used to perform the indicated tests. Differences in all comparisons were considered significant if the obtained $p$ values were less than 0.05.

**Reporting summary**. Further information on research design is available in the Nature Research Reporting Summary linked to this article.

## Data availability

The authors declare that the data that support the findings of this study are included in the published article and in the Supplemental Information, and are available from corresponding author upon reasonable request. Data containing RNAseq and RRBS raw sequencing read files were deposited onto sequencing read archive (SRA) with accession number PRJNA515359. The source data underlying Figs. 1b, 1c, 1e, 2b, 2d, 3b, 3d, and 4a–c and Supplementary Figs. 2b, 7b–c and 12c are provided as a Source Data file. New plasmids have been deposited to the Addgene repository.

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

## Acknowledgements

This work has been supported by the Jackson Laboratory internal grants (to A.C.), National Human Genome Research Institute 1R01HG009900 (to A.C) and National Cancer Institute P30CA034196 (to A.C), Leukemia Research Foundation New Investigator Grant (to S.L.), The Jackson Laboratory Director's Innovation fund 19000-17-31 (to S.L.), The Jackson Laboratory Cancer Center New Investigator Award (to S.L.), and National Cancer Institute of the National Institutes of Health P30CA034196 (to S.L.), and the National Institutes of Health grant CA115783 (to C.D.H.).

## Author contributions

A.T., C.H. and A.C. conceived and designed the study. A.T., N.J., M.D., A.R. performed the experiments. W.R. and S.L. analyzed RRBS data. A.T., C.H., and A.C. analyzed data and wrote the manuscripts.

## Competing interests

The authors declare no competing interests.
