## [Peer Review File · Nature Communications]

Reviewers' Comments:

Reviewer #1:

Remarks to the Author:

In this manuscript, Taghbalout et al describe the development of new DNA demethylation technique platforms (termed Casilio-MEs) based on the CRISPR/dCas9 targeting method, which achieves enhanced DNA demethylation and the activation of methylation-silenced genes. Although several other gene specific DNA demethylation methods have been reported, this work showed that the co-delivery of GADD45A or NEIL2 (proteins involved in the DNA repair pathways) with TET1 demethylase core domain improved the efficiency in DNA demethylation and the activation of MLH1 gene in HEK293T cells. Authors further showed that incorporation of DNA repair proteins in the new systems did not cause undesired mutations and the reported methods have comparable targeting specificity as previous methods. It was demonstrated that the reported platforms worked for 5 different target genes in 3 different cell lines. Overall, the manuscript is well written and the work provides a new, and potentially more effective, variation among other tools for DNA demethylation. However, there are several concerns that need to be addressed before the manuscript is suitable for publication in Nature Communications:

1. Several genome locus-specific (and simpler) DNA demethylation platforms already exist. The value of introducing and overexpressing additional key cellular regulatory proteins (GADD45A or NEIL2) is that it makes the new system more effective than previous ones. However, authors only demonstrated that this new system is more efficient than previous methods for one gene (MLH1). Considering the fact that authors found the Casilio-ME method had varied efficiency for different gene targets and may be locus dependent, it raises the question that if this new platform is universally more effective than previous methods as authors suggested, or may be more gene dependent that maybe at other loci, other methods may be equally or more effective? My concern is that the risk of overexpressing GADD45A or NEIL2 (see point 2 below) may not be warranted if the demethylation is not more efficient (in other gene loci). Authors may want to check a few other gene targets to see if the conclusion of Casilio-ME platforms being superior holds true to justify the co-expression of GADD45A or NEIL2.

2. Authors used full length GADD45A and NEIL2 in their new platforms instead of some core domains specific for the desired activities like the use of only the core domain of TET1. As authors mentioned that these proteins have chromatin or DNA-binding capability, and likely will interact with other cellular proteins. I wonder if overexpression of fusion proteins including full length GADD45A or NEIL2 may cause undesired cellular effects not related to the DNA demethylation at the target gene locus? Authors only examined the effect on MLH1 activation and showed that GADD45A or NEIL2 alone was not sufficient to induce MLH1 expression. I wonder if the authors have checked if the overexpression of GADD45A or NEIL2 in their systems changed the expression pattern of other genes (possibility of CRISPR/sgRNA unrelated "off-target" effects)?

3. Six sgRNAs were used simultaneously in the reported systems to guide the demethylation activity. Was the used of 6 sgRNAs an optimized result? Was it not as effective if using a single sgRNA? Since a more complex sgRNA structure was used (with multiple copies of binding sequences for PUFs), can author comment on how to decide where sgRNA target relative to the CpG islands to achieve the demethylation at desired loci? This will be a useful information for others when adapting this platform to other target genes.

4. Cells are harvested 3 days after transfection for subsequent assays, which seems to be long. Have authors done some time course experiments and found it took 3 day to give sufficient effects using the Casilio-ME system? A long editing period may allow additional secondary and downstream effects to occur that complicate the interpretation of observed results.

Reviewer #2:

Remarks to the Author:

The manuscript by Taghbalout, Cheng and colleagues explores optimization of RNA-guided targeted 5-methylcytosine (5mC) demethylation in DNA. Pathways for active DNA demethylation include the oxidation of 5mC by TET enzymes to generate 5hmC, 5fC and 5caC. These bases can either promote passive dilution of modified cytosines by excluding maintenance DNMT1 activity, or, for 5fC/5caC, be subject to active excision and replacement by unmodified cytosine. This later, active pathway involves the BER enzyme TDG and may be promoted by other factors including NEIL1/2 or GADD45A among others.

Recent work has shown that directing TET1 to a target locus can promote locus-specific demethylation. This can be achieved by fusion of TET1 to dCas9, with a sgRNA that directs the complex to the site of action. The gRNA can be further modified with additional RNA sequences (e.g. SunTag, MS2, etc.) that can bind to specific RNA binding protein to amplify the signal or to bring in other partners, which has been done with some success. Nonetheless, the poor overall efficiency of targeted DNA demethylation using existing scaffolds has limited their utility in published and unpublished data, and offers the impetus for this manuscript exploring experimental and conceptual changes to improve their activity.

In this manuscript the authors explore two added innovations in the framework of their Casilio-ME constructs to improve targeted DNA demethylation: (1) In the targeting RNA, they include various combinations/permutations of PUF-domain binding sequences (PBS) that can then be used to direct PUF domain fused proteins to a particular site; and (2) Using this approach, they then explore how combinations of additional players in the DNA demethylation pathway, beyond TET1, can be employed to improve activity.

Specifically, the authors demonstrate: (1) Using MLH1 as a representative locus, with multiple promoter-targeting sgRNAs, their ME1 construct results in gene induction, and the result is correlated with CpG demethylation by bisulfite. In direct comparisons to other systems of targeting TET or amplifying its localization with different RNA binding partners (SunTag, MS2) the PBS-based system showed superiority. (2) The delivery of GADD45A (via a PUF fusion) in a single fused construct or split construct (in ME2) leads to further enhancement in activation. (3) Co-delivery of NEIL2 (but not NEIL1, NEIL3 or TDG) also enhanced target gene activation in either fused or split constructs. (4) Using optimized systems in various permutations, the findings at the representative locus were extended to other exemplar loci, with the general conclusion that the coupled systems were typically more effective, albeit with site-to-site variability. (5) Sequencing supported the higher efficiency of demethylation with GADD45A or NEIL2 coupled systems, with possible trends in 5hmC at these sites that could point to mechanistic differences. (6) Epigenome editing appears to maintain the on-target versus off target balance observe other targeted epigenome editing methods as analyzed by RRBS and transcriptome profiling. (7) Importantly the authors mechanistically validate several aspect of their systems. For example, results were substantiated as involving the catalytic activity of TET1, and not due to non-catalytic TET activities or those of the other partner proteins (NEIL2 or GADD45A); with ME3 the result was validated as occurring due to NEIL2 localization and not simply its global overexpression.

Overall, I found this manuscript to be of significant impact and outstanding experimental rigor. The innovations employed are logical, and perhaps themselves not overwhelming, but such a rigorous approach to evaluating and improving an epigenome editing scaffold have been lacking and the systematic approach employed help to makes this work potentially more impactful. The most notable impact will be practical, offering an improved toolbox for epigenome editing. A secondary impact is that the observations here advance some interesting mechanistic questions as to how demethylation and gene activation are being achieved in these systems, which the authors start to address, but could be areas of further (future) inquiry.

The minor comments I have are directed towards a few points of clarification, and potential areas

where additional discussion could be beneficial. These points include:

1) In the experimental design, the authors employed multiple promoter-targeting sgRNAs in all settings. It is not clear if this is an obligate requirement of these systems to achieve efficient demethylation or whether there are dominant effects from single sgRNAs in the mix. The latter seems possible given the patterns of demethylation at MLH1, but this is not clear. Addressing this point better can impact how these tools will be employed.

2) The experimental data examining 5mC, 5hmC and C/5fC/5caC via BSeq and oxBSeq poses some challenges in interpretation. These data requires subtraction of two data sets to determine specific modification levels (BSeq-oxBSeq = 5hmC). The efficiency of 5hmC to 5fC conversion is critical in these methods. The authors ought to report highlight if the data from the two separate experiments were concordant (for example, were non-deaminated bases in BSeq < oxBSeq at every single C site) and report on any spike-in controls that confirm efficiency. In looking at the data, the very high levels of 5hmC at some sites is somewhat confusing and a bit surprising. For example, at site 250 with ME3.1 (Fig 4), 5hmC appears very high – perhaps more than 5mC. If real, “demethylation” in this system may often not mean conversion of 5mC to C, but rather changing 5mC to something else (in this case 5hmC). Clarifying the experiment to address the above questions about quality/quantification, and accounting for this different interpretation of what “demethylation” means at some loci could be of benefit to readers.

3) The observation that GADD45A and NEIL2 enhance activity when coupled to TET1 in these constructs begs several mechanistic questions which are only partially addressed. Some of the model employed is not that well-established. For example, there is evidence suggesting GADD45A enhancement which is cited, but other evidence that has failed to show an effect (e.g. Weber et al, Nat Comm, 2016). It would be fair to be even more clear in stating that the mechanisms by which these potential partners enhance demethylation are in need of further confirmation and study (p. 16, bottom).

4) I believe that there is no data directly proving that the demethylation observed at these loci is occurring via a TET-TDG dependent pathway, a point that is begged by the data showing that TDG itself does not couple with TET1 to increase targeted demethylation. It is worth considering and discussing the possibility that passive, replication-dependent demethylation could be occurring at these target loci. GADD45A or NEIL2 could stimulate TET1 activity at a locus to generate 5hmC (or 5fC, 5caC) which could then block maintenance DNMT1 activity and lead to promoter activation.

Response to reviewers' comments

Reviewer #1

(Remarks to the Author):

In this manuscript, Taghbalout et al describe the development of new DNA demethylation technique platforms (termed Casilio-MEs) based on the CRIPSR/dCas9 targeting method, which achieves enhanced DNA demethylation and the activation of methylation-silenced genes. Although several other gene specific DNA demethylation methods have been reported, this work showed that the co-delivery of GADD45A or NEIL2 (proteins involved in the DNA repair pathways) with TET1 demethylase core domain improved the efficiency in DNA demethylation and the activation of MLH1 gene in HEK293T cells. Authors further showed that incorporation of DNA repair proteins in the new systems did not cause undesired mutations and the reported methods have comparable targeting specificity as previous methods. It was demonstrated that the reported platforms worked for 5 different target genes in 3 different cell lines. Overall, the manuscript is well written and the work provides a new, and potentially more effective, variation among other tools for DNA demethylation. However, there are several concerns that need to be addressed before the manuscript is suitable for publication in Nature Communications:

1. Several genome locus-specific (and simpler) DNA demethylation platforms already exist. The value of introducing and overexpressing additional key cellular regulatory proteins (GADD45A or NEIL2) is that it makes the new system more effective than previous ones. However, authors only demonstrated that this new system is more efficient than previous methods for one gene (MLH1). Considering the fact that authors found the Casilio-ME method had varied efficiency for different gene targets and may be locus dependent, it raises the question that if this new platform is universally more effective than previous methods as authors suggested, or may be more gene dependent that maybe at other loci, other methods may be equally or more effective?

To address the raised question, we compared SunTag and dCas9-TET1(CD), which showed higher *MLH1* activation as compared to TALE-TET1 and MS2/dCas9-TET1, to *Casilio-ME1* in their efficiency to activate methylation-silenced genes other than *MLH1* in HEK293T (MGMT and RHOXF2) and LNCaP (GSTP1) cell lines and added the data to **Fig. S11 (panel i)** of the revised manuscript. This showed that *Casilio-ME1* is more effective than the reported TET1-targeting methods in activating the tested genes.

My concern is that the risk of overexpressing GADD45A or NEIL2 (see point 2 below) may not be warranted if the demethylation is not more efficient (in other gene loci). Authors may want to check a few other gene targets to see if the conclusion of Casilio-ME platforms being superior holds true to justify the co-expression of GADD45A or NEIL2.

To address the reviewer concern in gauging demethylation efficiency at other gene targets, we used *GSTP1* of LNCaP cells as example to ask whether the augmented gene activation obtained with *Casilio-ME3.2* came from an increased demethylation efficiency at the targeted CGI when compared to *Casilio-ME1*. High throughput BSeq showed that *Casilio-ME3.2* mediated an increased demethylation activity as expected from the increased gene activation obtained. We added the new data to **Fig. S11 (panels g and h)** of the revised manuscript.

2. Authors used full length GADD45A and NEIL2 in their new platforms instead of some core domains specific for the desired activities like the use of only the core domain of TET1. As authors mentioned that these proteins have chromatin or DNA-binding capability, and likely will interact with other cellular proteins. I wonder if overexpression of fusion proteins including full length GADD45A or NEIL2 may cause undesired cellular effects not related to the DNA demethylation at the target gene locus? Authors only examined the effect on MLH1 activation and showed that GADD45A or NEIL2 alone was not sufficient to induce MLH1 expression. I wonder if the authors have checked if the overexpression of GADD45A or NEIL2 in their systems changed the expression pattern of other genes (possibility of CRISPR/sgRNA unrelated “off-target” effects)?

The idea of using core domain instead of full-length protein is a good one, however GADD45A is a small protein (165 aa) made up of a single domain. For NEIL2 using catalytic domain without DNA-binding part of the protein might seem straightforward. The possibility that DNA-binding of these proteins might be a prerequisite for the obtained enhancement of gene activation cannot be excluded. Testing various truncations of these protein to identify the protein motif involved in enhancing TET1-mediated gene activation would be an area of future work to possibility obtain improved second generation of *Casilio-ME* platforms.

To address the raised points, we performed the following experiments:

1- We evaluated whether expression of GADD45A and NEIL2 as part of PUFa-TET1(CD) fusions could be toxic to transfected cells as alteration in expression of important gene(s) should impact cellular growth. MTT proliferation assay show no growth defect in *Casilio-ME* transfected cells when compared to control samples. We added data to **Fig. S10 (panel a)** of the revised manuscript.

2- We performed RNAseq analysis to compare transcriptome profiles of *Casilio-ME1* transfected cells to cells transfected with *Casilio-ME2.2* or *Casilio-ME3.1* to determine

alterations that might arise subsequently to GADD45A and NEIL2 *Casilio-ME* effectors expression. We added the data to **Fig. S10 (panel b)** of the revised manuscript. The RNAseq showed comparable RNA expression profiles with few off-target genes in *Casilio-ME2.2* transfected cells.

We followed up on the RNAseq analysis to validate four of the off-target deemed upregulated genes by RNAseq in *Casilio-ME2.2* cells. For these genes, we showed that expression of the GADD45A fusion protein in the absence of TET1 activity is not sufficient for the observed upregulation. We added data to **Fig. S10 (panel c)** of the revised manuscript.

3- We showed in the revised manuscript that expression of three genes (MGMT, SOX17 and RHOXF2) was essentially unchanged in cells transfected with *Casilio-ME1*, *ME2.2* and *ME3.1* effectors in the absence TET1 oxidative activity (dTET1(CD)) and CRISPR targeting (**Fig. S11f**).

3. Six sgRNAs were used simultaneously in the reported systems to guide the demethylation activity. Was the used of 6 sgRNAs an optimized result? Was it not as effective if using a single sgRNA? Since a more complex sgRNA structure was used (with multiple copies of binding sequences for PUFs), can author comment on how to decide where sgRNA target relative to the CpG islands to achieve the demethylation at desired loci? This will be a useful information for others when adapting this platform to other target genes.

To address the reviewer's comment regarding the selection of sgRNAs:

1- We added information on how sgRNAs for *MLH1* and other gene targets were chosen.

For *MLH1* target, we chose the sgRNAs based on efficiency of gene activation obtained with *Casilio-ME1* in the early stages of development. For the other gene targets (Fig. S11 a-e) no preliminary optimization was performed to select the sgRNAs, and the number of sgRNAs used varied (3 to 5 sgRNAs per CGI were used). So, sgRNA optimization is a plus but not required for the *Casilio-ME* system to work.

The sgRNAs used were designed randomly within the CGI targets and the design was oriented toward CRISPR-targeting efficiency. The idea was to use same set of sgRNAs to compare efficiencies across the platforms.

2- To address the point regarding sgRNA mixture effect:

2.1- We compared *Casilio-ME1* and SunTag systems in their efficiency to activate *MLH1* with one sgRNA. This showed that *Casilio-ME1* outperformed SunTag in activating *MLH1* expression. We added the data to **Fig. S1 (panel c)** of the revised manuscript.

2.2- Added data to **Fig. S6 (panel b)** from experiments using *Casilio-ME* platforms with one sgRNA. The obtained trend of augmented *MLH1* activation with six sgRNAs was recapitulated.

4. Cells are harvested 3 days after transfection for subsequent assays, which seems to be long. Have authors done some time course experiments and found it took 3 day to give sufficient effects using the Casilio-ME system? A long editing period may allow additional secondary and downstream effects to occur that complicate the interpretation of observed results.

During the early stages of the development of *Casilio-ME1* we have performed time-course experiments evaluating *MLH1* activation between day 3 and day 6 after transfection. We chose to collect cells on day 3 after transfection because other demethylation systems reported data from cells collected at day 3 or day 4, and we found that *MLH1* activation and CpG demethylation at Day 3 was slightly better than day 5 and 6. We then maintained cell collection time the same across the experiments for consistency.

Reviewer #2 (Remarks to the Author):

The manuscript by Taghbalout, Cheng and colleagues explores optimization of RNA-guided targeted 5-methylcytosine (5mC) demethylation in DNA. Pathways for active DNA demethylation include the oxidation of 5mC by TET enzymes to generate 5hmC, 5fC and 5caC. These bases can either promote passive dilution of modified cytosines by excluding maintenance DNMT1 activity, or, for 5fC/5caC, be subject to active excision and replacement by unmodified cytosine. This later, active pathway involves the BER enzyme TDG and may be promoted by other factors including NEIL1/2 or GADD45A among others.

Recent work has shown that directing TET1 to a target locus can promote locus-specific demethylation. This can be achieved by fusion of TET1 to dCas9, with a sgRNA that directs the complex to the site of action. The gRNA can be further modified with additional RNA sequences (e.g. SunTag, MS2, etc.) that can bind to specific RNA binding protein to amplify the signal or to bring in other partners, which has been done with some success. Nonetheless, the poor overall efficiency of targeted DNA demethylation using existing scaffolds has limited their utility in published and unpublished data, and offers the impetus for this manuscript exploring

experimental and conceptual changes to improve their activity.

In this manuscript the authors explore two added innovations in the framework of their Casilio-ME constructs to improve targeted DNA demethylation: (1) In the targeting RNA, they include various combinations/permutations of PUF-domain binding sequences (PBS) that can then be used to direct PUF domain fused proteins to a particular site; and (2) Using this approach, they then explore how combinations of additional players in the DNA demethylation pathway, beyond TET1, can be employed to improve activity.

Specifically, the authors demonstrate: (1) Using MLH1 as a representative locus, with multiple promoter-targeting sgRNAs, their ME1 construct results in gene induction, and the result is correlated with CpG demethylation by bisulfite. In direct comparisons to other systems of targeting TET or amplifying its localization with different RNA binding partners (SunTag, MS2) the PBS-based system showed superiority. (2) The delivery of GADD45A (via a PUF fusion) in a single fused construct or split construct (in ME2) leads to further enhancement in activation. (3) Co-delivery of NEIL2 (but not NEIL1, NEIL3 or TDG) also enhanced target gene activation in either fused or split constructs. (4) Using optimized systems in various permutations, the findings at the representative locus were extended to other exemplar loci, with the general conclusion that the coupled systems were typically more effective, albeit with site-to-site variability. (5) Sequencing supported the higher efficiency of demethylation with GADD45A or NEIL2 coupled systems, with possible trends in 5hmC at these sites that could point to mechanistic differences. (6) Epigenome editing appears to maintain the on-target versus off target balance observe other targeted epigenome editing methods as analyzed by RRBS and transcriptome profiling. (7) Importantly the authors mechanistically validate several aspect of their systems. For example, results were substantiated as involving the catalytic activity of TET1, and not due to non-catalytic TET activities or those of the other partner proteins (NEIL2 or GADD45A); with ME3 the result was validated as occurring due to NEIL2 localization and not simply its global overexpression.

Overall, I found this manuscript to be of significant impact and outstanding experimental rigor. The innovations employed are logical, and perhaps themselves not overwhelming, but such a rigorous approach to evaluating and improving an epigenome editing scaffold have been lacking and the systematic approach employed help to makes this work potentially more impactful. The most notable impact will be practical, offering an improved toolbox for epigenome editing. A secondary impact is that the observations here advance some interesting mechanistic questions as to how demethylation and gene activation are being achieved in these systems, which the authors start to address, but could be areas of further (future) inquiry.

The minor comments I have are directed towards a few points of clarification, and potential areas where additional discussion could be beneficial. These points include:

1) In the experimental design, the authors employed multiple promoter-targeting sgRNAs in all settings. It is not clear if this is an obligate requirement of these systems to achieve efficient

demethylation or whether there are dominant effects from single sgRNAs in the mix. The later seems possible given the patterns of demethylation at MLH1, but this is not clear. Addressing this point better can impact how these tools will be employed.

We have addressed the reviewer's comment regarding use of multiple sgRNAs by adding information and new data to the revised manuscript. As this point was also brought by reviewer #1, please see response to comment #3 above.

2) The experimental data examining 5mC, 5hmC and C/5fC/5caC via BSeq and oxBSeq poses some challenges in interpretation. These data requires subtraction of two data sets to determine specific modification levels (BSeq-oxBSeq = 5hmC). The efficiency of 5hmC to 5fC conversion is critical in these methods. The authors ought to report highlight if the data from the two separate experiments were concordant (for example, were non-deaminated bases in BSeq < oxBSeq at every single C site) and report on any spike-in controls that confirm efficiency. In looking at the data, the very high levels of 5hmC at some sites is somewhat confusing and a bit surprising. For example, at site 250 with ME3.1 (Fig 4), 5hmC appears very high – perhaps more than 5mC. If real, “demethylation” in this system may often not mean conversion of 5mC to C, but rather changing 5mC to something else (in this case 5hmC). Clarifying the experiment to address the above questions about quality/quantification, and accounting for this different interpretation of what “demethylation” means at some loci could be of benefit to readers.

In response to the referee's comment we added information to figure legend and Methods, and altered the relevant text to incorporate the suggested clarifications.

5hmC levels might appear surprisingly high in this case but the obtained results were reproducible and dCas9-TET1(CD) parallel control samples showed different 5hmC levels across the analyzed CpG. The finding here opens a framework to address mechanistic questions as how these different players contribute to the 5mC oxidative steps. The *Casilio-ME2* and *ME3* have the potential to serve such mechanistic studies as different combinations of protein assemblies that include mutants or other proteins could be tested.

3) The observation that GADD45A and NEIL2 enhance activity when coupled to TET1 in these constructs begs several mechanistic questions which are only partially addressed. Some of the model employed is not that well-established. For example, there is evidence suggesting GADD45A enhancement which is cited, but other evidence that has failed to show an effect (e.g. Weber et al, Nat Comm, 2016). It would be fair to be even more clear in stating that the mechanisms by which these potential partners enhance demethylation are in need of further confirmation and study (p. 16, bottom).

We agree and have added the citation to Discussion as suggested by the referee.

4) I believe that there is no data directly proving that the demethylation observed at these loci is occurring via a TET-TDG dependent pathway, a point that is begged by the data showing that TDG itself does not couple with TET1 to increase targeted demethylation. It is worth considering and discussing the possibility that passive, replication-dependent demethylation could be occurring at these target loci. GADD45A or NEIL2 could stimulate TET1 activity at a locus to generate 5hmC (or 5fC, 5caC) which could then block maintenance DNMT1 activity and lead to promoter activation.

As suggested by the referee's comment we have added alternative TDG-independent pathways to Discussion.

We thank the referees for their thorough evaluation of the manuscript and for their constructive comments.

Reviewers' Comments:

Reviewer #1:

Remarks to the Author:

All of my concerns have been addressed. I recommend to accept the manuscript for publication.

Reviewer #2:

Remarks to the Author:

The revised manuscript by Taghbalout, Cheng and colleagues focused on optimization of RNA-guided targeted 5-methylcytosine (5mC) demethylation in DNA has adequately addressed the minor points raised in the initial review. Specifically, the manuscript has been strengthened by the addition of comparisons to existing platforms at the request of Reviewer #1, more detailed discussion of the rationale for sgRNA selection at the request of both reviewers, and admission of some of the limitations of the mechanistic/models with regards to the role of NEIL2/GADD45 and the functionality of the TET/TDG pathway in these systems. The tools developed here should prove to be useful in an active and growing field focused on targeted epigenetic reprogramming.